# Salient alternatives facilitate implicatures

**Lewis Bott**[1]*, **Steven Frisson**[2]

**1** School of Psychology, Cardiff University, Cardiff, United Kingdom, **2** School of Psychology, Birmingham University, Birmingham, United Kingdom

\* Bottla@Cardiff.ac.uk

## Abstract

Sentences can be enriched by considering what the speaker does not say but could have done, the *alternative*. We conducted two experiments to test whether the salience of the alternative contributes to how people derive implicatures. Participants responded true or false to underinformative categorical sentences that involved quantifiers. Target sentences were sometimes preceded by the alternative and sometimes by a control sentence. When the target was preceded by the alternative, response times to implicature responses were faster than when preceded by the control sentence. This suggests that (1) alternative salience influences higher-level reasoning (2) the cost of deriving implicatures in sentence verification paradigms is due in part to low alternative salience.

**Data Availability Statement:** All data are available from the Open Science Framework database (DOI: 10.17605/OSF.IO/3BTYF).

**Funding:** The author received no specific funding for this work.

## Introduction

Listeners show an impressive ability to derive meaning beyond the lexical and compositional components of a sentence. What the speaker literally says conveys one message, but what the listener understands is something more. Grice [1] famously argued that such enrichments are a consequence of rational cooperation between interlocutors: Listeners consider not only what the speaker says, but also what they could have said but did not (the *alternatives*). For example, consider (1) below,

1. John ate some of the cookies.

⇒ John did not eat all of the cookies

2. John ate all of the cookies.

　A speaker that utters (1) might expect a listener to derive the inference that *John did not eat all of the cookies*. By reasoning that the speaker did not utter (2), the alternative, when it would have been more informative and relevant to do so, the listener can infer that (2) is not true. The resulting inference is known as a *quantity implicature* because it arises from Grice's quantity maxim (see [2–5] for more developed theories).

　Implicatures are optional enrichments and can be triggered by a range of factors. These include contexts where the upper bound of the quantifier domain is perceived to be relevant [1, 6]; socio-linguistic factors such as politeness [7]; and lexical markers of quantification (e.g. *of*; see [8]) and monotonicity (e.g. *if* or *any* [9]). In this paper we show that in addition to these factors, the psychological salience of the alternative is important: making people more aware

**Competing interests:** The authors have declared that no competing interests exist.

of the alternative makes it easier for them to derive the implicature. This suggests a model in which the activity level of the alternative is tied to implicature derivation.

## Alternative salience

Previous literature provides some support for the hypothesis that the salience of the alternative influences implicature derivation. In the developmental literature, a number of studies have argued that making the alternative more salient elevates the rate of implicatures in children (e.g.,[10, 11]). For example, Skordos and Papafragou [11] gave children an acceptability judgement task in which participants heard underinformative *some* sentences (e.g. "Some of the blickets have an X" when in fact all did) and manipulated when they were exposed to *all* sentences. When children heard the *all* sentences before the underinformative sentences–thereby making *all* accessible—rejection of the underinformative sentences was higher than when they heard them after. However, effects were not seen in adults, and so it is possible that they were due to developmental delays, such as the absence of an adult sense of relevance (e.g. [11]), rather than a fully developed pragmatic system.

Rees and Bott [12] tested the role of the alternative in adults. Participants completed a sentence-picture matching task with implicatures. They found that when the alternative was the prime to an ambiguous target trial involving a scalar expression, participants were more likely to derive an implicature interpretation for the target trial than when a literal interpretation was the prime. However, in a follow-up study, Marty et al. [13] argued that the alternative did not increase the rate of implicatures, but that instead, the literal prime lowered them (see [14]). Furthermore, these studies only reported choice proportions and did not measure the effect of the alternative on the time needed to derive the implicature.

Evidence that the alternative influences processing comes from visual world studies ([8, 15–17]). Huang and Snedeker [17] demonstrated that looks to an implicature target were delayed when targets were sometimes described with numbers, but not when they were only described with quantifiers. They argued that implicatures required a costly pragmatic inference in situations where the scalar trigger could not be lexically pre-loaded with the upper-bound meaning. Importantly, however, they were not testing whether alternative salience facilitated processing, only that the range of alternatives restricted the parser's ability to circumvent the enrichment process.

In summary, there is evidence that making alternatives more salient makes implicatures more likely, and that alternatives alter the strategies adopted by the processor. However, this evidence is either limited by subsequent studies or does not directly address the question of alternative salience. In this paper, we take one of the earliest paradigms that show delayed implicatures, Bott and Noveck [18], and demonstrate that the cost can be reduced when the context makes alternatives sufficiently salient.

## Experimental overview

Participants judged whether categorical sentences were true or false (Fig 1). Each sentence appeared on a separate screen. There were target sentences and prime sentences. Target sentences were underinformative *some* sentences that were true under a literal interpretation of *some* and false under an implicature interpretation, e.g. "some elephants are mammals" ([18]). Crucially, target sentences (underinformative sentences) immediately followed prime sentences (much like structural priming paradigms see e.g. [19, 20]). Sometimes prime sentences were alternatives, e.g. "all cows are mammals", and sometimes they were control sentences. If enrichment is sensitive to the salience of the alternative, implicature response times should be faster when the target is primed by the alternative than when primed by control sentences.

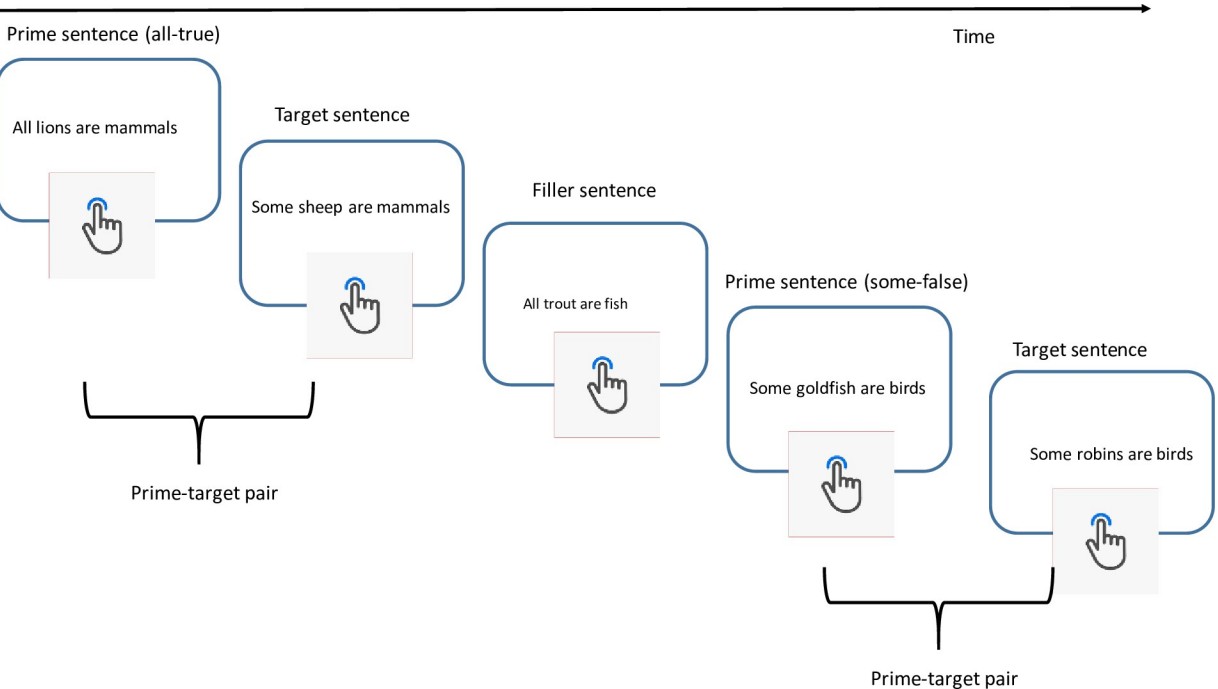

**Fig 1. Procedure.** Participants read a sentence and make a true/false judgement. Target sentences always appear after prime sentences. Prime-target pairs are interspersed with filler items.

We conducted two experiments. The procedure (Fig 1) and basic design was identical in both cases but the control conditions varied. Target sentences were underinformative sentences involving *some* (Table 1) and participants received feedback in a training phase to bias them towards implicature interpretations, i.e. that the expected answer was "false." In Experiment 1, prime trials were either *all*-true sentences, the *alternative*, or a control sentence, *all*-false or *some*-false. Sentences were designated alternatives according to the traditional Gricean approach to implicatures ([1, 21]): alternatives were sentences that were stronger than the target and relevant to the task. Consequently, *all*-true sentences were alternatives because they were stronger than the *some* target and because *all* was relevant (the quantifier needed to be processed to correctly judge the sentence as true or false). *Some*-false and *all*-false sentences were not alternatives because the quantifiers were not relevant (the sentence could be judged according to the subject-predicate relation only). *All*-false sentences were included to check

**Table 1. Stimuli.**

| Type | Name | Structure | Example | Correct | Count Exp 1 | Count Exp 2 |
|------|------|-----------|---------|---------|-------------|-------------|
| target | Under-informative | Some [exemplars] are [true superordinate] | Some elephants are mammals | F | 30 | 30 |
| prime/fillers | *some-false* | Some [exemplars] are [false superordinate] | Some goldfish are mammals | F | 10/20 | 10/10 |
| | *all-true* | All [exemplars] are [true superordinate] | All lions are mammals | T | 10/20 | 10/10 |
| | *all-false* | All [exemplars] are [false supordinate] | All goldfish are mammals | F | 10/20 | 0/10 |
| | *no-false* | No [exemplars] are [true superordinate] | No elephants are mammals | F | NA | 10/10 |
| filler | *no-true* | No [exemplars] are [false superordinate] | No elephants are cats | T | NA | 10 |
| | *some-true* | Some [exemplars] are [true subordinate] | Some elephants are Indian | T | 30 | 30 |

Note. Counts separated by "/" refer to prime/filler counts.

whether alternatives needed to be relevant or whether lexical activation of a potentially stronger quantifier would suffice (see [11]).

In Experiment 2 we again tested *all*-true and *some*-false sentences as primes but instead of *all*-false sentences we tested *no*-false sentences. *no*-false sentences were relevant to the target sentence, in that the quantifier needed to be processed to correctly judge the sentence, but were not classical alternatives to the target because *no* is weaker than *some*. Across the two experiments we therefore had an alternative prime, which was stronger than the target and relevant to the task, and controls representing stronger but not relevant (*all*-false), weaker but relevant (*no*-false), and neither stronger nor relevant (*some*-false).

## Method

### Participants

In each experiment, 38 Cardiff University students participated for course credit. Thirteen participants were assigned to two counterbalancing lists and 12 to the third. In Experiment 1, two participants were removed because they responded incorrectly for all prime trials, leaving 12 participants in each list. Ethical permission was granted by the Cardiff School of Psychology Ethics committee, EC.19.10.08.5703GA.

### Design and materials

Experimental sentences were primes or targets. Targets followed immediately after primes and were always underinformative *some* sentences (Table 1).

Primes were one of three types. In Experiment 1, primes were *all*-false; *all*-true; or *some*-false. In Experiment 2, primes were *all*-false; *no*-false; or *some*-false. There were 10 trials of each per participant. There were thus 30 pairs of experimental sentences. In addition, there were 90 fillers sentences in Experiment 1 and 80 in Experiment 2, distributed according to Table 1. These were included so that participants would not identify the prime-target structure (without fillers, an underinformative trial would appear every other trial) and to ensure that particular responses were not linked to particular quantifiers (e.g. without *no* filler items in Experiment 2, all sentences beginning with *no* would be false).

The experimental pairs appeared in a random order for each participant. Fillers were interspersed in a random order between pairs. There were no restrictions on the number of filler trials between experimental pairs. Thus there could be zero filler trials between one set of pairs, five fillers between another, 6 between another etc. depending on the random order assigned to each participant.

Items were constructed around 30 target sentences ("Some elephants are mammals") each with a different exemplar ("elephants"). For each, three prime sentences were created that shared the same superordinate category ("mammals") but involved different exemplars ("goldfish", "lions") and corresponded to the prime structures consistent with each condition (Table 1). The assignment of target sentence to condition was counter-balanced across three lists so that all target sentences appeared in all three conditions but no single participant saw the same target sentence more than once.

### Procedure

Participants pressed the "A" key for true responses and the "L" for false responses. They were given one example in the instructions, "All elephants are mammals," and told that this should receive a true response.

Participants underwent a training phase in which they judged 20 sentences. They received feedback on their responses, "Correct" or "Incorrect". This included four examples of the target sentences. Feedback encouraged an implicature response (false). They proceeded onto a testing phase in which they did not receive feedback.

One sentence was presented per trial. Sentences were presented in a single block in the centre of the screen. After participants pressed the key, the next sentence immediately appeared. There was no fixation cross or similar (this was deliberate to enhance the priming effect).

## Analysis

Data were analysed as mixed models with the lme4 package in R ([22]). The design was maximal ([23]) except that correlations between intercepts and slopes were suppressed to aid convergence (using lmer_alt(), [24]). Participants and items were included as random factors. All models converged. All data is available at DOI 10.17605/OSF.IO/3BTYF.

Main effects were established by comparing models with and without the prime factor using likelihood ratio tests. Simple effects p-values were computed with the Kenward-Roger and Satterthwaite approximations to degrees of freedom (lmerTest(), [25]).

## Results

Prime accuracy was high in Experiment 1, $M = 0.94$ (SD = 0.067) and Experiment 2, $M = 0.95$ (SD = 0.055) (Table 2). When analysing targets, responses to targets in which the prime was incorrect were removed, as is standard in structural priming paradigms (e.g. [19]).

In Experiment 1, accuracy on target sentences was high, $M = 0.76$, and there were no differences across prime (Table 2), $\chi^2(6) = 7.42$, $p = 0.28$. To analyse RTs to target sentences, we removed incorrect response to the target (24%) and RTs considered outliers (RT > 10s or RT < 100ms; N = 2 data points). There was a significant effect of prime on RTs (Fig 2), $\chi^2(6) = 24.65$, $p < .001$, such that all-true, $M = 1.6s$ (SD = 0.44), was significantly faster than some-false, $M = 1.8s$ (SD = 0.46), $\beta = -.14$, se = 0.031, $t = -4.31$, $p < .001$, and all-false, $M = 1.9s$ (SD = 0.50), $\beta = 0.16$, se = 0.033, $t = 4.90$, $p < .001$, but all-false did not differ significantly to some-false, $\beta = 0.027$, se = 0.031, $t = 0.88$, $p = 0.38$. Thus, response time for implicatures was significantly reduced when the alternative was relevant, but not when *all* was merely present in the prime.

A similar pattern was observed for Experiment 2. Accuracy to the targets was again high $M = 0.86$ and there were no differences across prime type, $\chi^2(6) = 3.10$, p = 0.80. Incorrect responses to the target (14%) and outliers (N = 5 data points) were removed for analysis of RTs. Significant effects of prime on RT to the target were observed (Fig 2), $\chi^2(6) = 16.97$, $p < .01$, such that the alternative, $M = 1.6s$ (SD = 0.36) was significantly faster than some-false, $M = 1.8s$ (SD = 0.47), $\beta = -0.094$, se = 0.030, $t = -3.14$, p < .01, and no-false, $M = 1.9s$

**Table 2. Prime and target response proportions.**

| Prime type | Experiment 1 | | Experiment 2 | |
|---|---|---|---|---|
| | Accuracy on prime | Accuracy on target subsequent to prime | Accuracy on prime | Accuracy on target subsequent to prime |
| *some-false* | 0.98 (0.063) | 0.76 (0.43) | 0.99 (0.043) | 0.86 (0.35) |
| *all-true* | 0.89 (0.14) | 0.78 (0.42) | 0.93 (0.098) | 0.86 (0.34) |
| *all-false* | 0.96 (0.088) | 0.75 (0.43) | NA | NA |
| *no-false* | NA | NA | 0.95 (0.098) | 0.87 (0.33) |

Note. Mean response proportions with standard deviations in parenthesis.

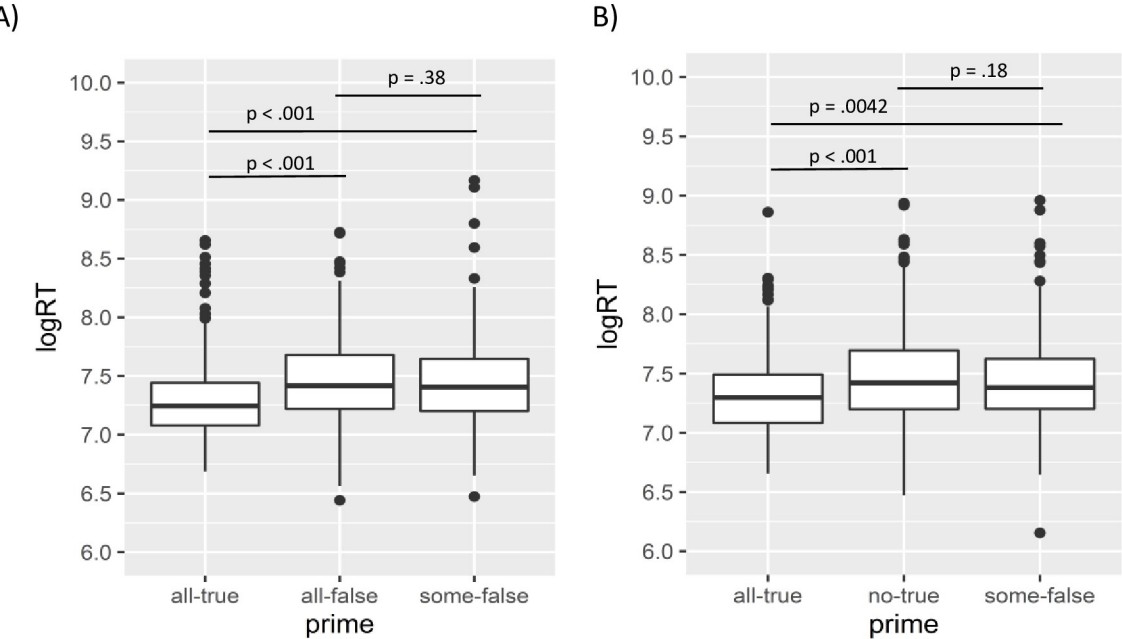

**Fig 2.** Box plots of logged response time to the target for (A) Experiment 1 and (B) Experiment 2. In both experiments response time was lower when preceded by the alternative (*all*-true) prime compared to either of the control sentences.

($SD = 0.51$), $\beta = -0.15$, se $= 0.033$, $t = -4.42$, $p < .001$, but some-false did not differ significantly to no-false, $\beta = 0.043$, se $= 0.032$, $t = 1.37$, $p = .18$. Thus, the reduction in processing time when *all* was salient was not due to the presence of any relevant quantifier, the quantifier needed to be stronger than the target.

## Discussion

We found that priming participants with the alternative (*all*-true sentences) speeded implicature responses. This was relative to primes that were neither relevant nor stronger than the target (*some*-false), relevant but not stronger (*no*-false), and not relevant but stronger (*all*-false). We next consider what might have caused this effect.

The alternative prime might have facilitated the construction of the alternative used by the target (the *target alternative*). There are two possibilities. The first is that the target alternative requires formulation, and this process was facilitated by the alternative prime. Formulating a sentence (e.g. [26]) requires selecting lexical expressions, identifying an appropriate syntactic frame, and mapping expressions to the frame, all of which require processing resources (e.g. [27]). The second is that conceptualization of the target alternative was made easier i.e. the prime helped identify which expressions were appropriate target alternatives. However, while both of these might have played a role in facilitating responses, neither seem likely to account for the 200ms priming effect we observed. The syntactic frame and lexical expressions used in the alternative and control primes were very similar, so there would have been little difference between help from the alternative prime and help from the control sentence. Likewise, while conceptualization of alternatives is generally a complex problem (see e.g. [4, 28]), there were few possible alternatives in our experiments, e.g. there were only two quantifiers (*some*, *all*) in Experiment 1. Participants would have been in no doubt as to which expressions were alternatives even after the control primes.

Instead, we suggest that the alternative prime influenced higher-level pragmatic processes, in particular the mechanism that triggers the implicature. At least two possibilities are consistent with our data. First, the activity of the alternative could be directly linked to the implicature enrichment mechanism. When the activity of the alternative exceeds threshold, the implicature could be triggered automatically ([12], [29]). After the alternative prime, the target alternative reached threshold more quickly than after control primes. Second, the salient alternative could have triggered recognition that the target sentence was underinformative. The comparison between *all*. . . and *some*. . ., and the recognition that *all*. . . was more informative than *some*. . ., might only start when *all*. . . was salient. This possibility is similar to Barner, Brooks and Bale's [30] suggestion that children are impaired on quantity implicatures because they are unable to recognize that *all*. . . is a more informative sentence than *some*. . . While children might fail to derive the implicature when the alternative is not salient, adults take more time to integrate the contextual cues but nonetheless derive the implicature.

### Literal or implicature priming?

In the Introduction we discussed Rees and Bott [12], in which participants were shown to derive an implicature interpretation more often after seeing an alternative prime than after a literal prime. Our findings are consistent with theirs but also extend their generality by showing that alternative priming effects are visible in response times, not just interpretations, and for linguistic stimuli, not just sentence picture combinations.

One argument against the claims of Rees and Bott [12] were that the literal sentences were priming participants to derive literal interpretations, rather than the alternative priming participants to derive implicatures ([13]). In other words, the default interpretation was the implicature and presentation of the alternative did not alter the default interpretation. Our task is less open to such an argument. The alternative prime speeded implicature responses relative to three other types of sentences (Fig 2) and while it is possible that each of three could have slowed target responses relative to a neutral baseline (instead of the alternative speeding responses relative to a baseline), we cannot see why that would be the case nor can we see what a more neutral baseline would be in our task than the three sentences we used.

### Differences with children

Our results are generally consistent with claims in the developmental literature about the importance of the alternative ([10], [11], [31]). Nonetheless, our findings differ to one of the most prominent studies, Skordos and Papafragou ([11]). Recall that Skordos and Papafragou found that introducing *all* sentences in a block prior to the *some* sentences elevated the rate of implicatures. Moreover, in Experiment 3, they found that *none* sentences had a similar effect. Skordos and Papafragou explained this by arguing that *none* was on the same scale as *all* and making the scale more salient also made *all* more salient. In contrast, we found that *all* sentences primed implicatures more than *no* sentences, and that there were no difference between *no* and the control.

The differing pattern could be because we tested adults whereas Skordos and Papafragou ([11]) tested children. Adults may have learned to quickly suppress irrelevant elements of a scale (so that *no* does not prime *all*) but not children. Another possibility is that there is something about linguistic stimuli, which we used, that makes it easier to inhibit elements of the scale, compared to graphical stimuli, as used by Skordos and Papafragou. Similarly, there may be differences in usage frequency between *no* and *none (of)* that contributes to the inhibition.

An alternative explanation relates to the focus placed on the quantifier in the respective tasks. In Skordos and Papafragou ([11]), correctly answering the *all* or the *no* trials meant

processing the quantifier and the predicate, whereas providing a true (literal) response to the *some* sentences required focussing only on the predicate. This meant that children who answered *all/no* sentences before *some* sentences were primed to focus on the quantifier when answering the *some* sentences whereas those who completed the *all/no* sentences after the *some* sentences were not primed. Skordos and Papafragou's finding that *all* and *no* influenced children's implicatures rates could therefore be because of quantifier priming rather than scale (alternative) priming. The reason why our results were different could be that in our study, sentences requiring quantifier and predicate processing were included throughout the task and so neither the *all* prime nor the *no* prime encouraged additional focus on the quantifier. Instead, the *all* sentences increased the salience of the alternative but the *no* sentences did not.

## Costs of implicatures

Previous research into implicatures has used processing cost (response time, choice proportions, eye fixations) to constrain mechanistic accounts of how implicatures are derived. For example, Bott and Noveck ([18]) found that implicature interpretations were delayed relative to literal meanings and consequently argued against a default implicature account. However, while some subsequent studies have confirmed that implicatures are delayed ([17, 32–38]), others have not ([8, 15, 39]), and there is no consensus about what causes the delay even when it is observed. Our research suggests that in sentence verification paradigms like this one, part of the reason that implicatures are costly is that the alternatives are not sufficiently salient. In other paradigms, the alternative may be more salient, thereby lowering the cost.

## Conclusion

Our data suggests that the salience of the alternative influences the derivation of implicatures: When alternatives are salient, implicatures are faster to derive. This constrains the range of processing models to those that assume alternative salience influences higher-level pragmatic reasoning. Furthermore, we have established that the cost of deriving implicatures in sentence verification paradigms is due in part to low alternative salience.

## Author Contributions

**Conceptualization:** Lewis Bott, Steven Frisson.

**Data curation:** Lewis Bott.

**Formal analysis:** Lewis Bott.

**Investigation:** Lewis Bott.

**Methodology:** Lewis Bott, Steven Frisson.

**Project administration:** Lewis Bott.

**Resources:** Lewis Bott.

**Writing – original draft:** Lewis Bott, Steven Frisson.

**Writing – review & editing:** Lewis Bott, Steven Frisson.

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
