## [Decision Letter · Decision Letter 0]

12 Jan 2022

PONE-D-21-38234Salient alternatives facilitate implicaturesPLOS ONE

Dear Dr. Bott,

Thank you for submitting your manuscript to PLOS ONE. After careful consideration, we feel that it has merit but does not fully meet PLOS ONE’s publication criteria as it currently stands. Therefore, we invite you to submit a revised version of the manuscript that addresses the points raised during the review process.

We look forward to receiving your revised manuscript.

Kind regards,

Andriy Myachykov, PhD

Academic Editor

PLOS ONE

Journal Requirements:

"NO authors have competing interests"

5. Please upload a copy of Figure 2, to which you refer in your text on page 6. If the figure is no longer to be included as part of the submission please remove all reference to it within the text.

6. Please ensure that you refer to Figure 1 in your text as, if accepted, production will need this reference to link the reader to the figure.

Reviewers' comments:

Reviewer's Responses to Questions

**Comments to the Author**

1. Is the manuscript technically sound, and do the data support the conclusions?

Reviewer #1: Yes

Reviewer #2: Yes

2. Has the statistical analysis been performed appropriately and rigorously? 

Reviewer #1: Yes

Reviewer #2: Yes

3. Have the authors made all data underlying the findings in their manuscript fully available?

Reviewer #1: Yes

Reviewer #2: Yes

4. Is the manuscript presented in an intelligible fashion and written in standard English?

Reviewer #1: Yes

Reviewer #2: Yes

5. Review Comments to the Author

Reviewer #1: The article describes two experiments which are aimed at investigating the role of alternative salience in scalar implicature derivation. Using a sentence verification task, they find that subjects are faster to respond to underinformative sentences containing the quantifier “some” (e.g., some elephants are mammals) when the sentences are preceded by an alternative as opposed to a control sentence. The alternative sentence consists of an “all” sentence that is true, e.g. all cows are mammals. (All elephants are mammals is the stronger alternative to the sentence Some elephants are mammals, and the prime uses the same structure.) Non-primes consist of false all-sentences (all goldfish are mammals) or false some-sentences (some goldfish are mammals).

The study forms a natural extension of related work: previous studies which have investigated the role of alternative salience used a picture-matching paradigm or were conducted with children. The authors also claim that they are first to look at reaction times, as opposed to just proportion of derived implicatures, in connection with alternative salience.

I find the paper easy to follow and the experimental design quite clever: the control sentences which serve as the RT baseline cover all other possibilities in terms of relevance and strength. The authors find that the priming effect emerges when the target is preceded by a true sentence with “all” but not when preceded by a false sentence with “all”, suggesting that the mere presence of the alternative quantifier does not facilitate implicature calculation but that the alternative also needs to be relevant.

I have several concerns about the article.

There was a practice phase at the start of the experiment which was meant to bias the participants to reject underinformative “some” sentences like “Some elephants are mammals”, i.e. to derive the scalar implicature. Later, they find that the “accuracy” on the target trials is 0.76 and 0.86 respectively for the two experiments, meaning that participants judged underinformative “some” sentences to be true some of the time. As far as I understand, the reaction times that are compared between conditions include the reaction times for those trials (on the other hand, reaction times for trials with wrong responses to the prime are excluded). However, since the question under investigation is whether alternative salience facilitates derivation of implicature, to me it doesn’t seem right to include the trials for which the implicature was not derived in the analysis. Does the effect the authors report persist once those trials are removed from analysis?

Participant exclusion criteria weren’t clear. The authors mention that “In Experiment 1, two participants were removed because they responded incorrectly for all prime trials, leaving 12 participants in each list.” Does this mean that if someone responded incorrectly to 90% of prime trials, their data is included? That seems quite liberal. Please report accuracy for each type of sentence.

The sample size is quite small, 38 participants per experiment. Given the fact that this is a relatively small effect (a 200ms speedup at the SD of 420-470ms for each condition), I would recommend the authors to collect more data to confirm that the effect persists, also when excluding the responses where the participants respond “True” to underinformative “some” sentences.

More minor concerns and questions:

In the Alternative salience section, the authors discuss the study by Rees and Bott (2018) which reports that the rate of implicatures is increased when the target is preceded by an alternative prime. Authors then also cite a follow-up study which came to a different conclusion: “in a follow-up study by Marty et al. (2021) argued that “However, in a follow-up study, Marty et al. (2021) argued that the alternative did not increase the rate of implicatures, but that instead, the literal prime lowered them (see also (Waldon & Degen, 2020).” They never return to this point - does their own experiment tease these two possibilities apart? I recommend elaborating on this point in the introduction and returning to it in the discussion. Also, the reference for Marty el at. (2021) is missing from the list of references.

“Responses to targets > 10s and < 100 ms were removed as outliers.” (p. 6) Why those numbers? 101ms seems quite short, too. Why not e.g. mean+3SD?

I would find it helpful if the study could also report the pairing of fillers, in order to better understand what the pattern was for whether true statements followed false ones etc. I could not determine from the description whether this was balanced.

In the Results for Experiment 1 (p. 6), the p-value is missing for one of the comparisons: “ all-false did not differ significantly to some-false, β = 0.027, se = 0.031, t < 1”.

In the Discussion, the authors state: “Our results are consistent with claims in the developmental literature about the importance of the alternative” but do not include any citations.

There are several points in the discussion where I find the phrasing too strong:

“The reason why our results were different is…” (p.8) this is an assumption, so rephase to something". According to this explanation, the reason ... would be"

“We have shown that the salience of the alternative influences the derivation of implicatures”. This is suggested by the experiment, not shown. What is shown is the difference in reaction times.

Other thoughts (neither positive nor negative)

This is just a thought, but in the part where the authors speculate about why they did not find a facilitatory effect with “No” whereas Skordos & Papafragou found one with “None”, could there be differences in processing “No” and “None (of)” that could underlie this difference? To me, “No X is a Y” seems to be a more rare construction, for instance.

Typos

p.7 this formulation process

p.7 the 200ms priming effect

p.7 the syntactic frame and lexical expressions used ... were very similar

p.7 something about linguistic stimuli, as we used, - something about the linguistic stimuli which we used?

p.8 whereas those completed -> whereas those who completed

p.8. raised the salience -> increased the salience?

Reviewer #2: PONE-D-21-38234

Summary: The paper presents the results of a study (2 slightly different versions) in which the presentation of a certain type of prime resulted in faster processing of an implicature. The different versions of the experiment were differences in the fillers trials and inclusion of no-false primes. Some evidence that alternative impacts implicature derivation in children, but there were clear differences in “no” in adults in the current study.

Evaluation: I have only minor suggestions for revision. Overall, I think the paper is a bit minimalist. � However, I think short, clear, and direct empirical studies are a good thing. I struggled somewhat with the paradigm, and so, why not include a figure in the methods or intro to clearly outline the task. I enjoyed reading the paper.

Minor Points:

1. In the abstract and introduction, argue that the alternative influences how implicature is derived. Later is says Cost=>ease vs. influence. The discussion is much clearer, particularly in the “Costs of Implicature” section. I think a bit of attention to detail in the earlier parts of the paper would enhance clarity.

2. Procedure is unclear about exact presentation of the prime and filler. Was the prime presented first? Or both together? I might have missed it, but it is clearly stated how the RT was calculated. Perhaps, it would be best to separate the design and materials sections.

3. Composition of fillers is not clear from the materials and Table 1. Presumably they included a prime and “target” sentence. It would be very strange for the fillers to consist of a single sentence. Also, how were they “interspersed” between criticals.

Really Minor Points:

1. Change General Discussion to Discussion

2. Subheading in the discussion S&P2016, seems a bit odd. Is Conflicting Findings or Differences with Children better?

6. PLOS authors have the option to publish the peer review history of their article (what does this mean?). If published, this will include your full peer review and any attached files.

Reviewer #1: No

Reviewer #2: No

---

## [Author Response · Author response to Decision Letter 0]

1 Feb 2022

Responses contained in a separate file.

---

## [Decision Letter · Decision Letter 1]

8 Mar 2022

Salient alternatives facilitate implicatures

PONE-D-21-38234R1

Dear Dr. Bott,

We’re pleased to inform you that your manuscript has been judged scientifically suitable for publication and will be formally accepted for publication once it meets all outstanding technical requirements.

Kind regards,

Andriy Myachykov, PhD

Academic Editor

PLOS ONE

Additional Editor Comments (optional):

Reviewers' comments:

Reviewer's Responses to Questions

**Comments to the Author**

1. If the authors have adequately addressed your comments raised in a previous round of review and you feel that this manuscript is now acceptable for publication, you may indicate that here to bypass the “Comments to the Author” section, enter your conflict of interest statement in the “Confidential to Editor” section, and submit your "Accept" recommendation.

Reviewer #1: All comments have been addressed

Reviewer #2: All comments have been addressed

2. Is the manuscript technically sound, and do the data support the conclusions?

Reviewer #1: Yes

Reviewer #2: Yes

3. Has the statistical analysis been performed appropriately and rigorously? 

Reviewer #1: Yes

Reviewer #2: Yes

4. Have the authors made all data underlying the findings in their manuscript fully available?

Reviewer #1: (No Response)

Reviewer #2: Yes

5. Is the manuscript presented in an intelligible fashion and written in standard English?

Reviewer #1: Yes

Reviewer #2: Yes

6. Review Comments to the Author

Reviewer #1: The authors have mostly addressed my concerns.

However, I believe there is still a spelling mistake in a name of an author in a reference. I think Tilly in Barr, Levy, Scheepers & Tilly, 2013 should be Tily. I already pointed this out previously. It is now corrected in the reference list, but not in the paper itself.

Reviewer #2: The authors have addressed all of my prior minor concerns and really minor concerns. Thus, I have no further comments.

7. PLOS authors have the option to publish the peer review history of their article (what does this mean?). If published, this will include your full peer review and any attached files.

Reviewer #1: No

Reviewer #2: No

---

## [Editor Report · Acceptance letter]

23 Mar 2022

PONE-D-21-38234R1 

Salient alternatives facilitate implicatures 

Dear Dr. Bott:

I'm pleased to inform you that your manuscript has been deemed suitable for publication in PLOS ONE. Congratulations! Your manuscript is now with our production department. 

Kind regards, 

on behalf of

Dr. Andriy Myachykov 

Academic Editor

PLOS ONE